# Body Fat and Muscle Mass in Association with Foot Structure in Adolescents: A Cross-Sectional Study

**DOI:** 10.3390/ijerph17030811

**Published:** 2020-01-28

**Authors:** Justyna Wyszyńska, Justyna Leszczak, Justyna Podgórska-Bednarz, Ewelina Czenczek-Lewandowska, Maciej Rachwał, Katarzyna Dereń, Joanna Baran, Justyna Drzał-Grabiec

**Affiliations:** Institute of Health Sciences, Medical College, University of Rzeszów, ul. Kopisto 2a, 35-959 Rzeszów, Poland; leszczakjustyna.ur@gmail.com (J.L.); j.e.podgorska@gmail.com (J.P.-B.); e.czenczek@univ.rzeszow.pl (E.C.-L.); maciekrachwal@gmail.com (M.R.); kderen@ur.edu.pl (K.D.); joannabaran.ur@gmail.com (J.B.); justyna.drzal.grabiec@wp.pl (J.D.-G.)

**Keywords:** adolescent, bioelectrical impedance, body composition, foot deformities

## Abstract

Prior studies have investigated associations between body mass index (BMI) and foot structure; however, these studies are limited only to the evaluation of the longitudinal arch of the foot and do not evaluate associations with body composition. Therefore, this study examined associations between body fat percentage (BFP) and muscle mass percentage with foot structure in adolescents. This study was conducted with 158 healthy subjects aged from 11 to 13 years. Body fat percentage and muscle mass percentage were estimated using bioelectrical impedance analysis. A podoscope was used to calculate Clarke’s angle (CL), the Wejsflog index (WI), hallux valgus angle (ALPHA), and the angle of the varus deformity of the fifth toe (BETA). Lower values of CL were found in participants with excessive BFP (*p* = 0.021). No differences were observed in the values of the Wejsflog, ALFA or BETA indices between normal and excessive BFP groups. Participants with the lowest muscle mass percentage were significantly more likely to have lower values of CL and WI (*p* = 0.014 and *p* < 0.001, respectively). Excess BFP appeared to have a significant effect on the longitudinal arch and low muscle mass percentage on the longitudinal and transverse arches of the foot in adolescents. There was no association between fat and muscle content with positions of the big and fifth toes.

## 1. Introduction

Over the past decades, the number of overweight and obese children has risen worldwide [1]. Orthopaedic complications related to obesity include musculoskeletal pain and discomfort, fractures, Blount’s disease, slipped capital femoral epiphysis, and both valgus and varus lower extremity misalignment [2]; however, the most frequent condition seems to be flatfoot [3].

Numerous studies have demonstrated that excessive body mass determined by body mass index (BMI), not body composition, has a negative effect on foot shape [4,5,6,7,8]. In contrast, other studies indicated that there is no association between increased body mass and foot posture in children [9,10,11]. Notably, individuals with increased BMI are not always obese in terms of body composition [12], and BMI is not a direct measure of body fat [13]. Individuals can have normal body weight and, at the same time, excess body fat [14].

Wearing et al. [15], in their study analysing the effect of body composition on footprint parameters in adults, found significant correlations between free fat mass and the area of both the hindfoot and forefoot as well as between fat mass and the midfoot area. The authors claimed that excessive fat mass may increase the midfoot contact area and, consequently, decrease the foot arch height, but they concluded that it is unclear if this represents a change in the osseous structure of the medial longitudinal arch. Riddiford-Harland et al. [16] used a pedograph to assess the footprint parameters in prepubescent children and found that children with a BMI greater than the 95th percentile had a decreased footprint angle and an increased Chippaux–Smirak Index which is characteristic of structural foot changes associated with compromised foot function. Dowling et al. [17] postulated that obese children are at an increased risk of developing foot discomfort and/or foot pathologies due to the increased plantar loads under the midfoot. However, it has been concluded that it is not clear if these changes are due to the presence of a fat pad in the midfoot region in obese children, causing a form of flatfoot. Mickle et al. [7] reported that flatfoot in overweight/obese preschool children was associated with differences in the osseous structure of the medial longitudinal arch.

Butterworth et al. [3], in their systematic review, concluded that obesity is strongly associated with flatfoot deformity, pronated dynamic foot function, and increased plantar pressures when walking. However, the authors indicated that there is limited evidence to support an association between other body composition measures, such as body fat mass, and foot structure. To the best of our knowledge, there is no study relating the association between muscle mass content and foot structure. Therefore, a comprehensive assessment of the association between body composition and footprint structure was necessary.

The aim of this study was to determine the association between body fat percentage (BFP) and muscle mass percentage with foot structure in adolescents aged from 11 to 13.

## 2. Materials and Methods

### 2.1. Participants

The study group consisted of students aged 11–13, who were enrolled in two schools in Rzeszów, Poland. The inclusion criteria: age between 11 and 13 years; absence of congenital disorders, neuromuscular problems, and foot deformations (confirmed on the basis of information from the parents and by our own observations); and consent to participate in the study.

The invitation to participate in the study was sent to all parents of children attending these schools (*n* = 940). The consent of 452 parents was obtained for child participation in this study. Of those respondents, 294 were excluded from the study for the following reasons: an age of less than 11 or greater than 13 years (*n* = 252), a functional state that did not allow for self-maintenance of a standing position (*n* = 2), previous orthopaedic surgery (*n* = 4), contraindications to perform bioelectrical impedance analysis (*n* = 2), taking medication affecting body composition (*n* = 3), refusal to participate in the study on the day of examinations (*n* = 6) and absence on the day of the examinations (*n* = 25). Ultimately, the study group consisted of 158 students (76 boys and 82 girls), aged 11–13 years.

### 2.2. Body Height, Body Weight and Body Composition

Participant body height was measured with accuracy to 1 mm using a portable stadiometer PORTSTAND 210 (Charder, Taichung, Taiwan ). The measurements were performed under standard conditions, with upright and straight body posture and bare feet.

Body fat percentage and muscle mass percentage were obtained using a Tanita device (BC-420) which uses foot-to-foot bioelectrical impedance analysis (BIA). The foot-to-foot method of BIA is a reliable and accurate tool for the measurement of body composition in the paediatric population [18]. Measurements were performed in the early morning after an overnight fast, because food or beverage consumption may decrease impedance by 4–15 Ω over a 2–4 hour period after meals representing an error of less than 3% [19]. Evaluation of body composition was performed according to standard research protocol. The participants were dressed in light clothing. The subjects’ feet were wiped with an alcohol pad. Once each surface dried, participants were instructed to stand upright on four electrodes with bare feet, with their arms abducted apart from their trunk and legs slightly spread [20,21]. Determinations of resistance and reactance were made by the BIA instrument, and BFP and muscle mass percentage were estimated from the manufacturer’s equations.

Differences in foot structure indices depending on BFP were assessed. For this purpose, BFP centile values by age and sex were used [22]. Two groups of subjects were distinguished: (i) normal BFP: <95th percentile of BFP (*n* = 136); (ii) excessive BFP: ≥95th percentile of BFP (*n* = 22).

We also assessed whether there were differences in the occurrence of foot disorders depending on muscle mass percentage. For this purpose, subjects were divided into 3 groups: (<Q1) the lowest quarter of measurements—subjects with the lowest muscle mass percentage (*n* = 39); (Q1–Q3) typical measurements (*n* = 80)—subjects between the upper and lower quartile; (>Q3) the highest quarter of measurements (*n* = 39)—subjects with the highest muscle mass percentage.

### 2.3. Foot Structure Indices

The CQ-ST podoscope was used as the main research tool. The podoscopic examination of the plantar side of the foot is a development and improvement on the well-known plantographic method. Photopodoscopy is a reliable quantitative analysis method which can be used in clinical and scientific analyses [23,24].

Both feet were subjected to examination and were scanned once in a standing position. Based on the picture obtained during the scan, footprint parameters were calculated for the dominant foot. The procedures for calculating the foot structure indices are shown in Figure 1 and Figure 2. The researcher marked by hand the appropriate points on the acquired footprint on the computer. Next, on the basis of those points, the computer calculated the indices describing the longitudinal and transverse arch of the foot and arrangement of the hallux and the fifth toe. All footprints were calculated by the same experienced investigator.

An example of a footprint scan computer image is presented in Figure 3.

The following parameters were measured [25,26]:Foot length (LG) is the line connecting the farthest points in the forefoot and the rearfoot (A–B), in mm (Figure 1).Foot width (WD) is the line connecting the metatarsale tibiale (C) and metatarsale fibulare (D) points, in mm (Figure 1).Clarke’s angle (CL) is located between the tangent to the medial edge of the foot (C-S) and the line joining the point of the largest recess (point Q) with the point of contact of the medial tangent to the forefoot (point q), in degrees. Clarke’s angle defines the medial longitudinal arch (Figure 2).The Wejsflog index (WI) is the ratio of the length to the width of the foot. The WI determines the transverse arch of the foot (Figure 1).Hallux valgus angle (ALPHA) is located between the tangent line to the medial edge of the foot (C–S) and the tangent to the pad of the big toe, derived from the metatarsale tibiale point (E–C), in degrees. The ALPHA defines the position of the big toe (Figure 1).The angle of the varus deformity of the fifth toe (BETA) is located between the tangent line to the lateral edge of the foot (D–T) and the tangent to the pad of the fifth toe, derived from the metatarsale fibulare point (F–D), in degrees. The BETA defines the position of the fifth toe) (Figure 1).

### 2.4. Statistical Analysis

Statistical analysis was performed using the SPSS Statistics 20 software. Variables were presented as mean, standard deviation, median, minimum, and maximum. The normality was tested by the Kolmogorov–Smirnov test, while the homogeneity of variance was assessed by Levene’s test. The independent *t*-test and one-way analysis of variance were used to examine the differences in quantitative variables. The level of statistical significance was adopted at *p* < 0.05.

### 2.5. Ethics

Before the study was initiated, written informed consent for participation was obtained from the children’s parents and study participants. The study was approved by the Bioethics Committee at the Medical Department of the University of Rzeszów (resolution number 7/05/2012) and conducted in accordance with ethical standards laid down in an appropriate version of the Declaration of Helsinki (as revised in Brazil 2013).

## 3. Results

General characteristics of the studied population are presented in Table 1. The average age of the subjects was 12.1 ± 0.88 yrs. The average height of the subjects was 150.3 ± 9.22 cm, while the mean body weight was 42.5 kg ± 10.59 kg and ranged from 22.4 kg to 79.8 kg. The average BFP was 18.5% ± 8.27% and ranged from 3.1% to 45.5%. The mean muscle mass was 32.2 kg ± 6.04 kg, while the muscle mass percentage was 77.2% ± 7.77% and ranged from 51.8% to 91.9%.

The differences in foot structure indices between healthy and excessive BFP groups are presented in Table 2. Lower values of CL were found in participants with excessive BFP (*p* = 0.021). These data indicate differences in the medial longitudinal arch of the foot among participants with excessive body fat content. No differences were observed in the values of Wejsflog, ALFA or BETA indices between healthy and excessive BFP groups.

The data collected in Table 3 show that in participants with the lower muscle mass percentage, the values of CL were significantly lower (*p* = 0.014). Subjects with the lowest muscle mass percentage (<Q1) were also significantly more likely to have lower values of WI than the subjects with the highest muscle mass percentage (>Q3) (2.58 versus 2.74, *p* < 0.001). These data indicate a decrease in the medial longitudinal arch and transverse arch of the foot in participants with lower muscle mass percentage.

## 4. Discussion

The findings of this study demonstrate that there was a significant association between BFP and muscle mass percentage with foot characteristics in adolescents. The mean value of CL among participants with excessive body fat was 12.25 degrees lower than in the group with normal body fat content. These data indicate a decreasing medial longitudinal arch among adolescents with excessive BFP. Lowering of the medial longitudinal arch and transverse arch of the foot were significantly more common among participants with the lowest content of muscle mass.

Woźnicka et al. [4] analysed the influence of BMI on the longitudinal arch of the feet among children aged between 3 and 13 years. On the basis of BMI, participants were divided into four subgroups: underweight, normal weight, overweight, and obese children. Comparing mean values of the CL in normal weight children with overweight and obese children, statistically significant differences were observed (*p* < 0.001). An increase of BMI values corresponded with average lower values of the CL characteristic of flat feet. The results of our research showed a similar relationship; however, the subjects were divided on the basis of BFP. The authors, who used the BIA to assess body fat content, also found positive linear associations between estimates of total body fat and midfoot contact area of footprints [15].

Excessive body mass results in greater skeletal overload and can therefore cause lowering of the longitudinal and transverse foot arch. Lowering of the foot arches can also be explained by overgrowth of adipose tissue, lower muscle strength [6,27] or by lowering of the bony structure of arches. However, Mickle et al. [7] indicated that children with excessive body mass do not have a thicker plantar pad, but they have a significantly lower plantar arch height compared to their non-overweight peers. Riddiford-Harland et al. [28] based on the results of ultrasound confirmed that the obese children (diagnosed on the basis of BMI) had not only significantly greater medial midfoot fat pad thickness relative to the leaner children but also lowered medial longitudinal arch height. Ezema et al. [29] also confirm that weight status was significantly associated with longitudinal flatfoot. Obese primary school children were three and a half times more likely to be diagnosed with flatfoot compared to children of normal weight. In another study, Riddiford-Harland et al. [16] suggested that excessive body weight seems to have a significant effect on the foot structure as early as at the age of 8 years. In prepubescent obese children who already manifest symptoms of foot deformities, problematic symptoms may develop later in life, especially if excessive body weight is maintained. It is, therefore, justified to undertake further research into the possible consequences, in particular any effects of excessive weight on foot functions.

The results of this study indicate that longitudinal and transverse foot arches are also associated with muscle mass percentage. The lower the muscle mass percentage in the examined adolescents, the lower the values of CL and WI observed which correspond to lowering both arches of the foot. This association can be explained by the hypothesis of poorer foot structure with lower muscle mass. However, data on this dependency are not clear. Golightly et al. [30] indicated that in adult individuals, foot symptoms and foot function may not be associated with leg muscle mass, but flat arch foot structure may be associated with higher muscle mass. The results of another study suggest that low muscle mass may be associated with high arch foot posture in the general community of older adults [31]. Muscle mass is potentially modifiable and could be considered as target for intervention to prevent or improve foot disorders. To gain a better understanding the aetiology of foot disorders, prospective studies evaluating the muscle mass specific to foot posture and function are needed. Future research should also investigate the long-term impacts of low muscle mass on foot structure and function in different populations.

We did not find any reports on the associations of body mass composition in adolescents on foot parameters such as the hallux or fifth toe setting. According to our knowledge, our results are the first concerning these associations. We have indicated that there are no statistically significant association between BFP and muscle mass percentage with digit deformities (ALFA, BETA). However, there are reports claiming that, in adulthood, excessive weight contributes to the development of hallux varus, hallux valgus, digitus quintus varus, widening of the heel area, and heel misalignment [3,6,27,32]. Considering the fact that adolescents with excessive adipose tissue often remain obese in adulthood, musculoskeletal complications of childhood obesity may aggravate at a later age. Therefore, promoting health and preventing obesity in addition to diagnosing and correcting abnormalities in foot posture in the population of children and adolescents is essential.

Investigating associations between body mass composition and foot disorders in a population of early adolescence is preliminary in this topic. Our findings need to be continued. Large-scale research should be carried out taking into account a larger number of children of all ages.

## 5. Limitations

The large sample size in this study allowed for assessment of an association between a subject’s body mass composition and footprint-based parameters; however, it was not large enough to carry out logistic regression analysis. Therefore, it is essential to undertake further studies to confirm current study findings in a larger sample size of children and adolescents in different age groups. This will allow for additional stratification into subgroups according to age or gender and allow for a more thorough investigation of the influence of different confounding factors on footprint-based parameters.

This paper reports the association of body composition on static foot posture. Future research should assess dynamic foot function and prevalence of foot symptoms.

In spite of the limitations, we believe that the results of this study will contribute new information and knowledge to public health.

## 6. Conclusions

Adolescents with excessive BFP had significantly lowered medial longitudinal arches of the foot compared to peers with healthy fat content. Adolescents with low muscle mass percentages were predisposed to decreasing medial longitudinal arches and transverse arches of the foot. There was no association between fat and muscle content with positions of the big and fifth toes.

## Figures and Tables

**Figure 1 ijerph-17-00811-f001:**
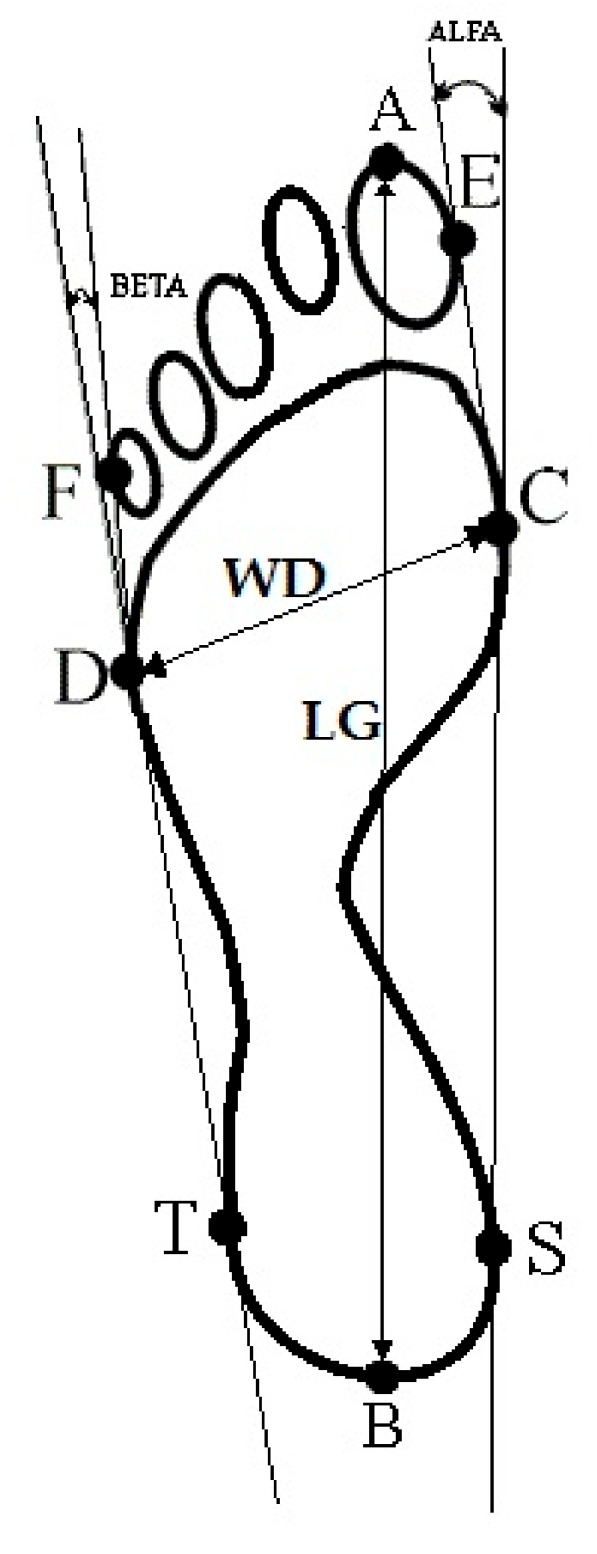
Procedure for determining the feet structure indices (i.e., foot length, foot width, hallux valgus angle, the angle of the varus deformity of the fifth toe, and the Wejsflog index).

**Figure 2 ijerph-17-00811-f002:**
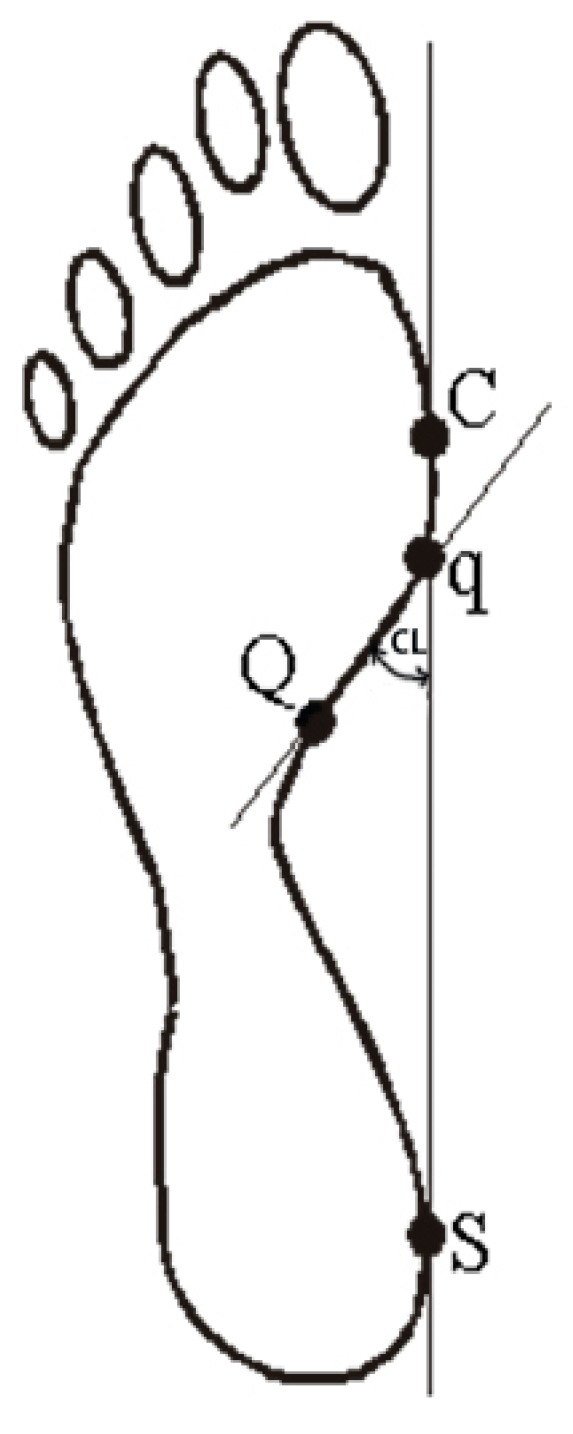
Procedure for determining Clarke’s angle.

**Figure 3 ijerph-17-00811-f003:**
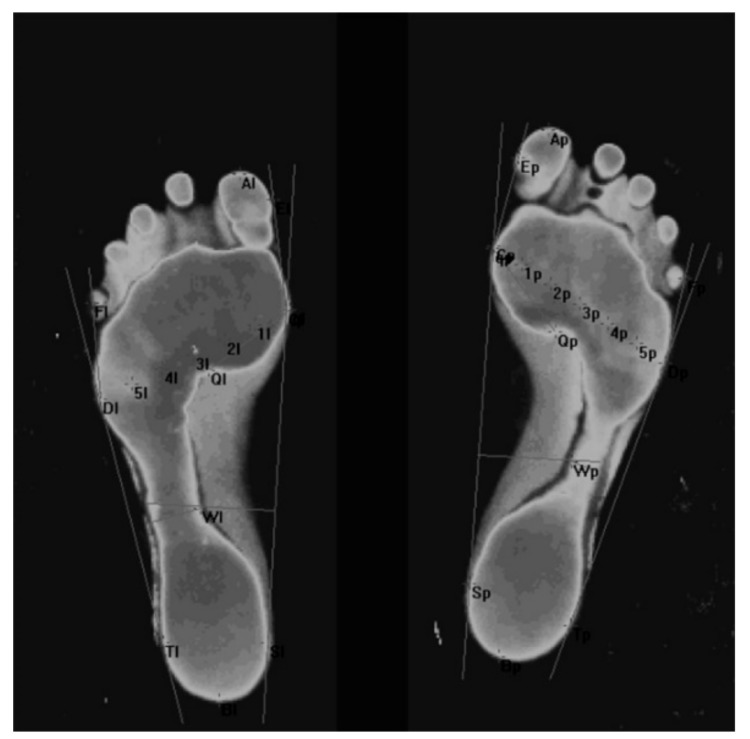
Example of a footprint scan computer image.

**Table 1 ijerph-17-00811-t001:** Anthropometric parameters and body mass composition.

	Mean	Median	SD	Minimum	Maximum
Age (years)	12.13	12.00	0.88	11.00	13.00
Body height (cm)	150.30	149.00	9.22	129.00	172.00
Body mass (kg)	42.48	41.25	10.59	22.40	79.80
BMI (kg/m^2^)	18.63	18.05	3.52	11.90	33.60
BFP (%)	18.52	16.63	8.27	3.13	45.49
Fat mass (kg)	8.48	6.75	5.81	1.00	36.30
Muscle mass (kg)	32.23	31.55	6.04	19.50	48.80
Muscle mass (%)	77.22	78.90	7.77	51.75	91.88

BFP—body fat percentage; BMI—body mass index.

**Table 2 ijerph-17-00811-t002:** Assessment of foot structure indices and body fat percentage.

Foot Structure Parameter	Body Fat Percentage (BFP)				*p*-Value
	Normal		Excessive		
	Mean	SD	Mean	SD	
Clarke’s angle (°)	43.83	10.99	31.58	15.61	0.021
Wejsflog index	2.70	0.18	2.61	0.18	0.098
ALFA (°)	5.22	3.58	5.21	3.89	0.992
BETA (°)	9.79	6.30	10.13	7.14	0.862

ALFA—hallux valgus angle, BETA—the angle of the varus deformity of the fifth toe. *p*-Values were calculated using the independent *t*-test.

**Table 3 ijerph-17-00811-t003:** Assessment of foot structure indices and muscle mass percentage.

Foot Structure Parameter	Muscle mass percentage						*p*-Value
	<Q1		Q1–Q3		>Q3		
	Mean	SD	Mean	SD	Mean	SD	
Clarke’s angle (°)	39.25	13.46	42.69	10.04	46.98	12.39	0.014
Wejsflog index	2.58	0.16	2.73	0.16	2.74	0.17	<0.001
ALFA (°)	6.10	4.01	4.68	3.42	5.44	3.38	0.115
BETA (°)	10.56	6.15	9.53	6.45	9.65	6.42	0.698

ALFA—hallux valgus angle, BETA—the angle of the varus deformity of the fifth toe, Q—quartile. *p*-Values were calculated using the one-way analysis of variance (ANOVA).

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
