# Peer review of "Body Fat and Muscle Mass in Association with Foot Structure in Adolescents: A Cross-Sectional Study"

_ijerph, 2020, doi:10.3390/ijerph17030811_

Round 1

Reviewer 1 Report

The topic of the  association between body fat and muscle mass with foot structure in adolescents is interesting. The manuscript is good, but it still requires corrections:

The Introduction should be slightly changed. Too much space has been devoted to describing the known connections between obesity and various diseases. Too little space was devoted to the description of the effect of excessive body fat and too much muscle mass on foot structure.  A part of text  from line 39 to line 54 could be shortened and part of text  from line 55 to line 61 could be the extended. I do not understand why it was assumed that excessive fatness was 25% and 30% of body weight in boys and girls, respectively. The fatness will change with age, especially during puberty. In my opinion the 95th percentile should have been used as the cut-off point (McCarthy HD et al. 2006. Body fat reference curves for children. Int J Obes (Lond). 30(4):598-602).

Best regards!

Author Response

Dear Reviewer,

We are very thankful for your comments about our article “Body fat and muscle mass in association with foot structure in adolescents: a cross-sectional study”. We included all your helpful advice and made necessary changes in the article.

Comments 1. The topic of the  association between body fat and muscle mass with foot structure in adolescents is interesting. The manuscript is good, but it still requires corrections:

The Introduction should be slightly changed. Too much space has been devoted to describing the known connections between obesity and various diseases. Too little space was devoted to the description of the effect of excessive body fat and too much muscle mass on foot structure.  A part of text  from line 39 to line 54 could be shortened and part of text  from line 55 to line 61 could be the extended.

Response 1: According to the reviewer's suggestion, the introduction has been redrafted.

Comments 2. I do not understand why it was assumed that excessive fatness was 25% and 30% of body weight in boys and girls, respectively. The fatness will change with age, especially during puberty. In my opinion the 95th percentile should have been used as the cut-off point (McCarthy HD et al. 2006. Body fat reference curves for children. Int J Obes (Lond). 30(4):598-602).

Best regards!

Response 2. We agree with the reviewer's opinion. The division of children due to the cut-off points we proposed was not adequate. Therefore, we used body fat% centile values by exact age, and we used the 95th percentile as the cut-off point of excessive fatness (McCarthy HD et al. 2006. Body fat reference curves for children. Int J Obes (Lond). 30 (4): 598-602). The Methods section and the Results section have been corrected. We have performed a new statistical analysis of the results, which are presented in Table 2.

The language correction of the article has been made by a native speaker.

Again, we appreciate all your insightful comments and the opportunity to revise our paper. Thank you for taking the time and energy to help us improve the paper. We hope our revision meets your approval.

Yours faithfully,

Authors

Author Response

Dear Reviewer,

We are very thankful for your comments about our article “Body fat and muscle mass in association with foot structure in adolescents: a cross-sectional study”. We included all your helpful advice and made necessary changes in the article.

Comments 1.

- Title: The title is correct as it reflects correctly the objective and hypothesis of the work.

- Summary: This section follow a well structured format.

-  Keywords; Please use recognised MeSH terms as this will assist others when they are searching for information on your research topic. The following website will provide these (simply start typing in a keyword and see if it exists or find an alternative if it does not): https://www.ncbi.nlm.nih.gov/mesh

Response 1. Thank you for suggestion. Keywords have been rewritten according to MeSH terms.

Comments 2.

- Introduction: The research question its not important and the subject has been previously developed with greater scientific rigor.

Response 2.

The introduction has been redrafted. Space devoted to describing the connections between obesity and various diseases has been replaced by the description of the effect of body composition on foot structure.

To the best of our knowledge, there is no study relating the association between muscle mass content and foot structure. According to our knowledge, our results are the first concerning these associations.

In our study we used two main research tools: the CQ-ST podoscope (description in Response 3) and body composition analyser which uses bioelectrical impedance analysis (BIA). Considerable research has gone into developing BIA monitors that can distinguish between lean and fat tissue on the basis of their differential conductance and impedance characteristics. These techniques are slightly less accurate than the more sophisticated research tools, but offer an important practical advantage in being simple and cheap to use.

BIA monitors has been validated against DXA in mixed populations. Results show strong significant correlations between both methods for fat mass, % body fat, and total free fat mass:

-Karelis AD, Chamberland G, Aubertin-Leheudre M, Duval C; Ecological mobility in Aging and Parkinson (EMAP) group. Validation of a portable bioelectrical impedance analyzer for the assessment of body composition. Appl Physiol Nutr Metab. 2013; 38(1): 27-32;

- Pietrobelli A, Rubiano F, St-Onge MP, Heymsfield SB . New bioimpedance analysis system: improved phenotyping with whole-body analysis. Eur J Clin Nutr. 2004; 58: 1479–1484;

 -Achamrah N, Colange G, Delay J,et al. Comparison of body composition assessment by DXA and BIA according to the body mass index: A retrospective study on 3655 measures. PLoS One. 2018;13(7):e0200465.

Comments 3.

- Materials and Methods: computerized platforms are used today, the method is really old and its scientific validity is not proven.

Response 3.

The CQ-ST podoscope was used in our study. The tool provides developed and refined podoscope assessment. Apart from the image (plantoconturogram), the equipment provides information on the spatial structure of the foot. The results obtained are repeatable and comparable. Photopodoscopy is a reliable quantitative analysis method which can be used in clinical and scientific analyses [Pita-Fernández S1, González-Martín C, Seoane-Pillado T, López-Calviño B, Pértega-Díaz S, Gil-Guillén V. Validity of footprint analysis to determine flatfoot using clinical diagnosis as the gold standard in a random sample aged 40 years and older. J Epidemiol. 2015;25(2):148-54].

Moreover, there are many studies published in international journals, in which the authors used the same device as we do, for example: Puszczalowska-Lizis E, Bujas P, Omorczyk J, Jandzis S, Zak M. Feet deformities are correlated with impaired balance and postural stability in seniors over 75. PLoS One. 2017; 12(9): e0183227.

Comments 4.

- References: Appropriate

Again, we appreciate all your insightful comments and the opportunity to revise our paper. Thank you for taking the time and energy to help us improve the paper. We hope our revision meets your approval.

Yours faithfully,

Authors

Reviewer 3 Report

Dear Authors

congratulations on an innovative approach to the topic. I am very interested in the problem of posture defects in children, including foot defects, and your manuscript made me very interested. I have some suggestions for the presented results, which I posted in the form of comments in the attached file.
I wish you further success in scientific work.

Author Response

Dear Reviewer,

We are very thankful for your comments about our article “Body fat and muscle mass in association with foot structure in adolescents: a cross-sectional study”. We included all your helpful advice and made necessary changes in the article.

Congratulations on an innovative approach to the topic. I am very interested in the problem of posture defects in children, including foot defects, and your manuscript made me very interested. I have some suggestions for the presented results, which I posted in the form of comments in the attached file.

I wish you further success in scientific work.

Comments 1. Statistical analysis. Have the distribution normality and uniformity of variance been studied? What tests?

Response 1. Yes, the distribution normality and uniformity of variance have been studied. The normality was tested by the Kolmogorov-Smirnov test, while the homogeneity of variance was assessed by Levene's test.

Comments 2. Ethics. Participants or their parents or legal guardians signed an informed consent? Participants were underage.

Response 2. Before the study was initiated, written informed consent for participation was obtained from children’s parents and study participants.

Comments 3. please show two decimal places as for other data in table 1 first column.

Response 3. The data was supplemented as suggested by the reviewer.

Comments 4. Table 2. How many participants had normal and how many excessive BFP? Was it equal for girls and boys? Maybe gender was important here?

Response 4. In total participants with excessive BFP were 22 (10 boys and 12). Gender have no effect on the results.

Comments 5. Table 3. How many participants were in each of these three groups? This is very important information, but it has not been provided. Is it possible to additionally describe how many participants with low muscle mass had high BFP at the same time?

Response 5. Participants with the lowest muscle mass percentage (<Q1) were 39; with typical measurements (Q1-Q3) were 80, and with the highest muscle mass percentage (>Q3) were 39.

There were 19 participants with the lowest muscle mass percentage and excessive BFP at the same time (and 3 participant with typical measurements regarding muscle mass percentage and excessive BFP at the same time).

Comments 6. Discussion, line 183: higher or lower values of CL?

Response 6. Thank you for noticing the error. It is corrected on “lower”.

Again, we appreciate all your insightful comments and the opportunity to revise our paper. Thank you for taking the time and energy to help us improve the paper. We hope our revision meets your approval.

Yours faithfully,

Authors

Reviewer 4 Report

Introduction: Authors have highlighted the importance of obesity and its associations with foot posture - but fail to mention why these foot posture issues are important? Further details required concerning symptoms i.e. pain and burden i.e. disability, mobility issues.

Methods: - Please provide resolution number of the Bioethics Committee.

-The authors should provide detailed inclusion and exclusion criteria for the sample.

-Was the standard research protocol for BIA an international standard?

Methods/Results: It seems to that it would be better to divide the subjects body fat content using body fat % centile values by exact age.

Discussion: -The authors claim that the results of this study conclude that foot overload may lead to the development of foot disorders. However this was a cross-sectional study and therefore causation can not be extrapolated.

-The authors mention that the results of similar studies could be used to improve prophylaxis of foot deformities. There was no mention of pain or disability measurement in this study, indeed for the latter subjects with disability were excluded. So this suggests that these deformities were not problematic and therefore do not require intervention.

Author Response

Dear Reviewer,

We are very thankful for your comments about our article “Body fat and muscle mass in association with foot structure in adolescents: a cross-sectional study”. We included all your helpful advice and made necessary changes in the article.

Comments 1. Introduction: Authors have highlighted the importance of obesity and its associations with foot posture - but fail to mention why these foot posture issues are important? Further details required concerning symptoms i.e. pain and burden i.e. disability, mobility issues.

Response 1.  According to the reviewer's suggestion, the introduction has been redrafted.

Comments 2. Methods: Please provide resolution number of the Bioethics Committee.

Response 1.  Resolution number of the Bioethics Committee has been provided.

Comments 3. The authors should provide detailed inclusion and exclusion criteria for the sample.

Response 3.  The above data has been provided.

Comments 4. Was the standard research protocol for BIA an international standard?

Response 4. Yes. Evaluation of body composition was performed according to standard research protocol [Kyle, U.G., Bosaeus, I., Lorenzo, A.D., Deuenberg, P., Elia, M., Goméz, J.L., et al. Bioelectrical impedance analysis-part II: review of principles and methods. Clin. Nutr. 2004, 23, 1430–1453.]

Comments 5. Methods/Results: It seems to that it would be better to divide the subjects body fat content using body fat % centile values by exact age.

Response 5. We agree with the reviewer's opinion. The division of children due to the cut-off points we proposed was not adequate. Therefore, we used body fat% centile values by exact age, and we used the 95th percentile as the cut-off point of excessive fatness (McCarthy HD et al. 2006. Body fat reference curves for children. Int J Obes (Lond). 30 (4 ): 598-602). The Methods section and the Results section have been improved. We have performed a new statistical analysis of the results, which are presented in Table 2.

Comments 6. Discussion: The authors claim that the results of this study conclude that foot overload may lead to the development of foot disorders. However this was a cross-sectional study and therefore causation can not be extrapolated.

Response 6. We agree with the reviewer. This sentence has been deleted.

Comments 7. The authors mention that the results of similar studies could be used to improve prophylaxis of foot deformities. There was no mention of pain or disability measurement in this study, indeed for the latter subjects with disability were excluded. So this suggests that these deformities were not problematic and therefore do not require intervention.

Response 6. We agree with the reviewer. This sentence has been reworded.

Again, we appreciate all your insightful comments and the opportunity to revise our paper. Thank you for taking the time and energy to help us improve the paper. We hope our revision meets your approval.

Yours faithfully,

Authors

Round 2

Reviewer 2 Report

I still is not very original, the photos of the footprint plattform are not very good and the determination of the plantar arch is unrealiable.